# Position: A Theory of Deep Learning Must Include Compositional Sparsity

**David A. Danhofer** [* 1 2]  **Davide D'Ascenzo** [* 1 3 4]  **Rafael Dubach** [* 1 5]  **Tomaso Poggio** [* 1]

## Abstract

Overparametrized Deep Neural Networks (DNNs) have demonstrated remarkable success in a wide variety of domains too high-dimensional for classical shallow networks subject to the *curse of dimensionality*. However, open questions about fundamental principles, that govern the learning dynamics of DNNs, remain. In this position paper we argue that it is the ability of DNNs to exploit the compositionally sparse structure of the target function driving their success. As such, DNNs can leverage the property that most practically relevant functions can be composed from a small set of constituent functions, each of which relies only on a low-dimensional subset of all inputs. We show that this property is shared by all efficiently Turing-computable functions and is therefore highly likely present in all current learning problems. While some promising theoretical insights on questions concerned with approximation and generalization exist in the setting of compositionally sparse functions, several important questions on the learnability and optimization of DNNs remain. Completing the picture of the role of compositional sparsity in deep learning is essential to a comprehensive theory of artificial – and even general – intelligence.

## 1. Introduction

Deep Neural Networks (DNNs) have achieved remarkable breakthroughs across numerous domains, including computer vision, playing games (Silver et al., 2016a), protein structure prediction (Jumper et al., 2021; Abramson et al., 2024), natural language usage, and complex reasoning (Romera-Paredes et al., 2023; Trinh et al., 2024;

DeepSeek-AI et al., 2025). Despite their rapidly growing set of achievements, our understanding of DNNs still lags behind their empirical success. Without deeper theoretical insights, it remains unclear why certain architectures scale so well to high-dimensional tasks, or how to pinpoint the limits of Deep Learning (DL) paradigms.

Historically, many attempts to build intelligent systems relied on logical or rule-based approaches (Newell & Simon, 1976; Hayes-Roth et al., 1983). In contrast, the rise of DNNs from around 2012 onward ushered in learning-based architectures that surpass traditional algorithms in numerous domains. For several years, the focus has been on pushing performance boundaries: from high-dimensional image recognition (AlexNet (Krizhevsky et al., 2012), ResNet (He et al., 2015)) to playing strategic games (AlphaGo (Silver et al., 2016b), AlphaZero (Silver et al., 2018)). More recently, Large Language Models (LLMs) such as the GPT models (Brown et al., 2020; OpenAI (2023), 2023) have advanced natural language understanding and generation. Although these models continue to break benchmark after benchmark, their development often outpaces the foundational theory needed to explain or predict their performance.

Despite these empirical advances, research in DL theory is striving to keep pace, aiming to address three foundational questions:

1. *Approximation*: How well can Neural Networks (NNs) approximate functions of interest?

2. *Optimization*: How can we optimize NN parameters on finite training data?

3. *Generalization*: Why are NNs able to avoid overfitting the training data despite their large capacity?

Building on Poggio & Fraser (2024) and related work (Mhaskar et al., 2016; Poggio et al., 2017; Bauer & Kohler, 2019; Schmidt-Hieber, 2020), **we argue compositional sparsity – a property shared by all efficiently Turing-computable functions – explains how DNNs can represent, learn and generalize in high-dimensional settings without suffering from the curse of dimensionality**. Importantly, the curse of dimensionality manifests itself in two distinct ways: it encompasses both the exponential

*Equal contribution [1]Center for Brains, Minds and Machines (CBMM), MIT, Cambridge, MA, USA [2]ETH Zurich, Zurich, Switzerland [3]Politecnico di Torino, Torino, Italy [4]University of Milan, Milan, Italy [5]University of Zurich, Zurich, Switzerland. Correspondence to: Davide D'Ascenzo <davide.dascenzo@unimi.it>.

*Proceedings of the 42nd International Conference on Machine Learning*, Vancouver, Canada. PMLR 267, 2025. Copyright 2025 by the author(s).

growth in the number of parameters required to approximate high-dimensional functions (the approximation aspect), and the exponential increase in sample complexity and poor convergence rates encountered during training (the optimization aspect). While the theory of compositional sparsity provides a compelling explanation for how deep networks can overcome the curse of dimensionality in terms of approximation, our understanding of how these networks efficiently optimize or learn such representations remains comparatively underdeveloped. In practice, providing partial structural hints often makes the learning problem much more manageable. For example, exposing intermediate steps (as in chain-of-thought prompting) or restricting layers to operate on only a small subset of inputs (as in convolutional architectures) can effectively guide the model toward the correct hierarchical decomposition. This in turn can reduce optimization complexity and improve the interpretability and generalization of trained networks.

We begin by introducing the classical learning framework and the curse of dimensionality, which bars shallow learners from approximating complex, high-dimensional functions with polynomially many parameters. Next, we contrast this view with deep learners, who may leverage the compositional sparsity of functions to avoid the curse of dimensionality in approximation. Subsequently, we shift our view to optimization and generalization contrasting existing findings and open questions. We conclude our work by presenting alternative views to our proposed angle and summarize our claims.

## 2. Classical Learning and the Curse of Dimensionality

In the following, we briefly introduce statistical learning viewed through the lens of Empirical Risk Minimization (ERM) following standard literature (Hastie et al., 2009; Vapnik, 2013; James et al., 2023). We also introduce the notion of the *curse of dimensionality*, to which classical learners such as shallow networks are subject.

In a supervised learning problem, an unknown probability measure $\mu$ in the space of input-output pairs $\mathcal{X} \times \mathcal{Y}$ gives rise to the target function $f_\mu \colon \mathcal{X} \to \mathcal{Y}$ assumed to underlie an observable i.i.d. finite training set $S = \{(x_i, y_i)\}_{i=1}^m$ with $y_i = f_\mu(x_i)$. It is the goal to approximate $f_\mu$ using some $f$ best, as measured by an expected risk over the probability measure $\mu$

$$\mathcal{R}(f) \;=\; \int \ell(f(x), y) \, d\mu(x, y) \qquad (1)$$

constructed from some per-instance loss $\ell$. Since $\mu$ is commonly unknown, the expected risk is approximated empiri-

cally via the observable training set,

$$\mathcal{R}_{\mathrm{emp}}(f) \;=\; \frac{1}{m} \sum_{i=1}^m \ell(f(x_i), y_i). \qquad (2)$$

A low empirical risk does not necessarily imply that the approximation predicts well on unseen data. However, the generalization performance of this ERM procedure can be bounded by taking the complexity of the hypothesis class $\mathcal{H}$, over which $f$ is optimized, into account. The Rademacher complexity $\mathcal{R}_m$ of $\mathcal{H}$, defined on the independent Rademacher random variables $\sigma_i$,

$$\mathcal{R}_m(\mathcal{H}) = \mathbb{E}_\sigma \left[ \sup_{f \in \mathcal{H}} \frac{1}{m} \sum_{i=1}^m \sigma_i f(x_i) \right] \qquad (3)$$

probabilistically bounds the gap between the expected risk $\mathcal{R}(f)$ and the empirical risk $\mathcal{R}_{\mathrm{emp}}$

$$\mathcal{R}(f) \leq \mathcal{R}_{\mathrm{emp}}(f) + \mathcal{R}_m(\mathcal{H}_f) + o(\delta) \qquad (4)$$

where the low-order error terms depend on the certainty of the guarantee $1 - \delta$ and the training set size $m$.

From an approximation-theoretic perspective, the key challenge is that classical learners such as kernel machines or shallow networks are affected by the curse of dimensionality. As the dimension $d$ of a function $f$ grows, these methods may require an exponentially large number of parameters to approximate $f$ arbitrarily well (Poggio et al., 2017).

Consequently, a central question arises: *Which property of real-world target functions ensures that a suitable DNN can approximate them without requiring a number of parameters that grows exponentially with $d$?* The crucial insight is that deep networks can exploit suitable structural assumptions about the target function—most notably, that it admits a decomposition which avoids the need for an exponential number of parameters. As we will demonstrate in the following section, this requirement turns out to be surprisingly mild encompassing a broad class of functions. Thus, while traditional approaches suffer in high dimensions, DNNs can circumvent an exponential blow-up in the number of parameters by leveraging such structure (Beneventano et al., 2021).

## 3. Compositional Sparsity and Deep Learning

A central concept in our argument is that of *efficient Turing-computability*. Throughout this work, we use this term to refer to functions in the complexity class **FP**, that is, function problems solvable by a deterministic Turing machine in polynomial time. While the decision problem class **P** involves single-bit yes/no answers, **FP** encompasses functions whose outputs can be computed in polynomial time. For real-valued functions, computability can be formalized

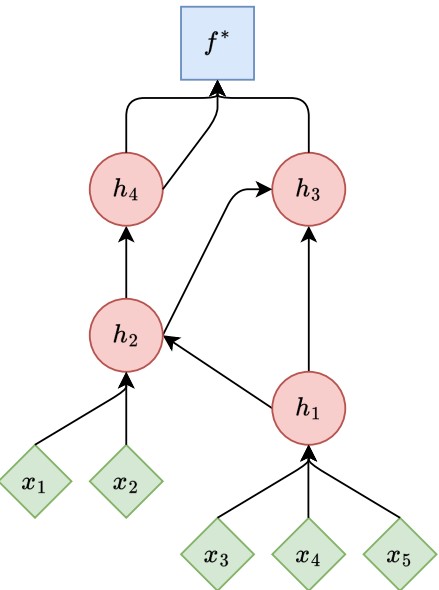

*Figure 1.* A DAG representing a compositionally sparse function. The green diamonds denote the $d = 5$ input variables, the red dots constituent functions and the blue square the final output. Each function depends on at most $3 = c \ll d$ variables.

in multiple ways (see Poggio & Fraser 2024 for a detailed discussion), but for the purposes of this work, we equate "efficient Turing-computable functions" with those in **FP**.

Subsequently, we introduce the notion of compositionally sparse functions and their relevance. We then state how this property helps DNNs to break the curse of dimensionality. For an alternative set of proofs see (Poggio, 2025).

### 3.1. Compositionally Sparse Functions

**Definition 3.1** (Compositionally Sparse Function). A function $f : \mathcal{X}^d \to \mathcal{X}$ is *compositionally sparse* if it can be represented as the composition of at most $\mathcal{O}(\text{poly } d)$ constituent functions each of which is sparse, i.e., depends on at most a (small) constant number of variables $c$.

A compositionally sparse function can be visually depicted as a Directed Acyclic Graph (DAG), in which the leaves represent inputs, the root denotes the output function, and the internal nodes represent the constituent functions. The maximum in-degree of the DAG is equivalent to $c$. Figure 1 illustrates this definition for a small toy example with total input dimension $d = 5$, and input dimension of at most $c = 3$ for all constituent functions. Functions with constituents of bounded dimensionality were earlier referred to as "hierarchically local compositional functions" (Poggio et al., 2017). Interestingly, this property is not very restrictive and is satisfied by a large class of functions, as the following theorem states.

**Theorem 3.2** (Efficient Computability implies Compositional Sparsity, restated from Poggio & Fraser 2024). *Any function that is efficiently Turing-computable is compositionally sparse.*

We prove this conjecture from Poggio & Fraser 2024. The proof uses the fact that efficiently Turing-computable functions may be translated into Boolean circuits of a polynomial number of gates with a bounded number of input variables. The full proof is deferred to Appendix A.

### 3.2. Deep Learning under Compositional Sparsity

Compositionally sparse functions may be approximated by suitable deep networks while avoiding the curse of dimensionality—this is generally not the case for shallow networks.

**Theorem 3.3** (Poggio et al. 2017, informal). *Suppose $f$ is a compositionally sparse function of input dimension $d$ with DAG $G_f$. Let the complexity of a network be the number of its trainable parameters. Then*

1. ***Shallow Networks** with one hidden layer require complexity $\mathcal{O}(\epsilon^{-d})$ to approximate $f$ to an accuracy $\epsilon > 0$, which is the best bound possible, whereas*

2. ***Deep Networks** mimicking $G_f$ require complexity $\mathcal{O}(d\epsilon^{-2})$ to approximate $f$ to an accuracy $\epsilon > 0$.*

Combining Theorem 3.2 on the compositional property of all efficiently computable functions with Theorem 3.3 from above, yields the following corollary:

**Corollary 3.4** (cf. Poggio & Fraser (2024)). *Any efficiently Turing-computable function (Boolean or real-valued) may be approximated to an accuracy of $\epsilon > 0$ with a deep, sparse network of polynomial complexity in $d$ and $\epsilon^{-1}$.*

As such, DNNs are universal approximators for all practically computable functions, capable of avoiding the curse of dimensionality. Observe, that the argument in the case of approximation only requires the in-degree of the function nodes in the DAG to be constrained, but does not further restrict the function class from which the "node functions" are drawn.

## 4. Learnability and Optimization of Compositionally Sparse Functions

It is noteworthy, that the curse of dimensionality occurs not only in the context of network complexity, i.e., a question related to approximation. It also extends to optimization, namely, convergence rates and sample complexity. While compositional sparsity explains why deep networks can *represent* efficiently computable functions with polynomial

complexity, it does not guarantee that such representations can be *learned* efficiently from input-output pairs. This section explores the theoretical and practical challenges of learning compositionally sparse functions and connects these insights to modern training paradigms.

## 4.1. Theoretical Challenges

Learning general compositionally sparse functions from input-output examples faces fundamental complexity barriers. A foundational result by Goldreich et al. (1986) demonstrates that, under standard cryptographic assumptions (specifically, the existence of one-way functions), there exist families of Boolean circuits of polynomial size that are not efficiently learnable in the distribution-free setting by any polynomial-time algorithm, regardless of the representation used. This result is representation-independent and applies directly to the class of functions computable by polynomial-size Boolean circuits, which encompasses compositionally sparse functions as defined in this work.

**Theorem 4.1** (Goldreich et al. 1986, informal). *Assuming the existence of one-way functions, there exists a polynomial $p$ such that the class of Boolean circuits with at most $p(n)$ gates is not learnable in polynomial time by any polynomial-time evaluable representation class.*

This cryptographic hardness result implies that, without additional structural assumptions or access to more than just input-output pairs, learning arbitrary compositionally sparse functions is infeasible in the worst case. In other words, even though such functions can be efficiently represented and computed, their learnability from examples alone is fundamentally limited by computational complexity barriers.

However, this worst-case hardness does not preclude the efficient learnability of many important subclasses of compositionally sparse functions. For example, Mansour (1994) showed that most sparse Boolean functions are easy to learn, and further work has identified tractable subclasses such as staircase functions (Abbe et al., 2021) or sparse polynomials (Negahban & Shah, 2012). The key insight is that while compositional sparsity is necessary for efficient representation, it is not sufficient for efficient learnability in the absence of further assumptions or structural information. In practice, providing partial supervision, architectural biases, or exploiting additional properties of the constituent functions can make learning feasible.

Despite these theoretical barriers, DNNs that likely represent compositionally sparse functions often achieve impressive performance on real-world tasks with high-dimensional data. This empirical success suggests that, in practice, the structure of real-world problems and the inductive biases of DNNs frequently circumvent worst-case hardness. Some

theoretical results support this phenomenon: E.g., Bauer & Kohler (2019) show that in non-parametric regression, the convergence rate of shallow networks diminishes exponentially with the input dimension, whereas DNNs can achieve convergence rates that depend only on the interaction order of the underlying compositional function, not the ambient dimension. Further work (Schmidt-Hieber, 2020; Kohler & Langer, 2021; Dahmen, 2023; Cagnetta et al., 2024) demonstrates that DNNs can overcome the curse of dimensionality and efficiently learn compositional functions under suitable assumptions.

An interesting example of sparse Boolean functions is provided by staircase functions, which are hierarchically structured Boolean functions over a high-dimensional hypercube. Such functions can be learned in polynomial time using layer-wise stochastic gradient descent on specialized DNNs (Abbe et al., 2021). The hierarchical structure enables gradient descent to progressively combine low-level features into higher-level ones through network depth. Negahban & Shah (2012) show that functions representable as $s$-sparse Boolean polynomials over $n$ variables can be learned via $L_1$-constrained convex optimization inspired by compressed sensing, achieving $O(s^2 n)$ sample complexity. Similarly, Boolean functions with Fourier spectra concentrated on $k$ non-zero coefficients (out of a predefined set $P$ of potential basis elements) can be learned via $L_1$-constrained regression with $O(k \log^4 |P|)$ samples (Stobbe & Krause, 2012).

These results collectively highlight that, although the worst-case learnability of compositionally sparse functions is computationally hard, many natural and structured subclasses remain efficiently learnable. This dichotomy underscores the importance of leveraging additional structure – whether through architectural design, training paradigms, or supervision – to circumvent the limitations imposed by general hardness results.

## 4.2. Implications for Architecture Design

Convolutional Neural Networks (CNNs) address compositional sparsity not because of translational invariance but because the filters are constrained to local patches, which induces sparse Toeplitz weight matrices. This architectural bias, which is independent of the convolutional property, ensures that the network computes a compositionally sparse function. Recent work by Xu et al. (2023) demonstrates that such sparsity can be leveraged to obtain significantly tighter generalization bounds than those derived from naive Rademacher complexity, for both sparse and dense networks. This suggests that, in domains where CNNs excel – that is, where the underlying function is well-approximated by a compositionally sparse structure – deep sparse learners are essential for strong generalization. Notably, these generalization results depend on the sparsity of the weight matrices,

not on convolution per se.

On the optimization side, compositional sparsity also impacts the symmetry properties of the network. Dense architectures, such as fully connected networks, possess large symmetry groups: permutations of neurons or layers often leave the function unchanged, resulting in highly degenerate loss landscapes with many equivalent minima (Kawaguchi, 2016; Brea & Gerstner, 2019; Nguyen & Hein, 2017; Gissin et al., 2019). In contrast, compositionally sparse architectures – including CNNs – have much smaller symmetry groups, as their constrained connectivity restricts the set of permissible permutations. For example, CNNs avoid permutation symmetries in weight space by enforcing translational equivariance, which simplifies the optimization landscape and can accelerate convergence. Recent works have begun to explore general principles of learning in- and equivariances in DNNs (van der Ouderaa & van der Wilk, 2022) and the interplay between sparsity, equivariance, and symmetry reduction (Ziyin, 2024), suggesting that architectural biases that reduce or exploit symmetry can induce useful structure and constraints on learning, thereby serving as a general mechanism for improving learnability and optimization.

While the structural sparsity induced by CNNs has empirically been shown to be an exceptional fit for vision tasks (Yamins et al., 2014; Cichy et al., 2016; Eickenberg et al., 2017), it can generally be challenging to impose the correct sparse structure *a priori* in architectures intended for other domains. Considering the natural language domain, it is reasonable to assume that the entities sharing a meaningful relationship in a sentence or text may shift position and have variable relative pair-wise distances. The rigid sparse structure of a CNN may therefore not be well-suited. Transformers, on the other hand, may address this by dynamically learning to *focus* on a still small, but flexible input-dependent subset of tokens via attention (Vaswani et al., 2017; Han et al., 2023; Poggio, 2023). Song et al. (2025) rigorously characterize how sparse attention mechanisms in transformers approximate exact attention, revealing that attention patterns naturally exhibit sparsity despite the architecture being dense by design. Recent empirical studies have further shown that transformers often exhibit emergent compositional structure across layers (Murty et al., 2022; Vig & Belinkov, 2019; Petty et al., 2023).

We posit that the key lies in the autoregressive training framework: **by training models to predict each token conditioned on previously generated tokens, autoregressive training encourages the incremental construction of complex outputs from simpler components. This process can naturally lead to the emergence of compositional sparsity through learned intermediate representations**.

## 4.3. Universality of Auto-Regressive Predictors and Chain-of-Thought

A recent result by Malach (2023) can be derived as a corollary of Theorem 3.2. Malach's theorem – obtained independently and in a somewhat different context –establishes that autoregressive next-token predictors are universal learners for any efficiently Turing computable function. Specifically, Malach (2023) states:

**Theorem 4.2** (Malach 2023, informal). *For any function $f$ that is efficiently Turing-computable, there exists a dataset $\mathcal{D}$ such that training a (linear) auto-regressive next-token predictor on $\mathcal{D}$ results in a predictor that approximates $f$.*

Because Boolean sparse functions are easy to learn (Mansour, 1994, Theorem 4.9), it is easy to show that our Theorem 3.2 implies the following statement (which is effectively equivalent to Theorem 4.2):

**Corollary 4.3** (informal). *Any function $f$ that is efficiently Turing-computable, can be learned if training sets are available for each of the sparse constituent functions in one of its decompositions.*

Interestingly, decision trees (Mansour, 1994, Theorem 5.10) are sparse Boolean functions because they have small $L_1$-norm. In a smoothed complexity setting where the input distribution is drawn from the biased binary hypercube, and the bias of each variable is randomly chosen, the class of $\log(n)$-depth decision trees satisfies the general staircase property. In turn, staircase functions with a small number of terms are sparse, since they are a sum of parity functions of increasing degree.

Poggio & Fraser (2024) argue that a dataset containing the step-by-step output and therefore the intermediate results of constituent functions should suffice to learn any compositionally sparse function conditioned on the learnability of its constituents. The structure of natural language is commonly regarded to be sparse s.t. subsequent tokens only depend on few prior tokens (Liu et al., 2022; Murty et al., 2022). This compositional structure can be picked up by auto-regressive next-token predictors (Gan et al., 2024). The vast corpora of natural language (Radford et al., 2019; Raffel et al., 2020), on which modern-day LLMs are trained on, are therefore natural candidates for such datasets, that contain not only end-to-end examples of input-output pairs of functions, but also intermediate results. This may explain complex reasoning observed in present-day LLMs.

Recent empirical work by Lindsey et al. (2025) provides direct evidence that large language models, such as Claude 3.5 Haiku, internally perform genuine multi-step reasoning in practice. For example, when prompted with *Fact: the capital of the state containing Dallas is*, the model produces the correct answer "Austin" by first inferring that Dallas is in Texas, and then that the capital of Texas is Austin.

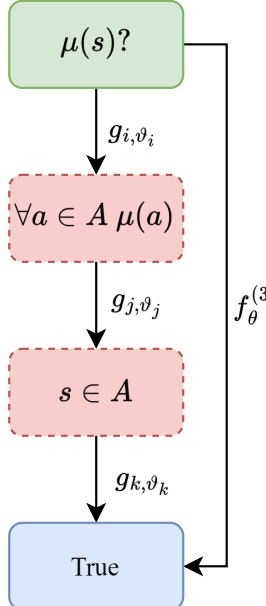

*Figure 2. Is Socrates mortal?* CoT-style intermediate solving steps can simplify this famous question to a sequence of general reasoning steps of less complexity than the specific question at hand.

Attribution graph analysis reveals that the model's computation proceeds through distinct intermediate representations corresponding to "Texas" and "capital," rather than relying solely on memorized shortcuts. This multi-hop reasoning is reflected in the model's internal feature activations and their interactions, supporting the view that compositional sparsity and hierarchical reasoning are not only theoretically motivated but also empirically realized in state-of-the-art models (Lindsey et al., 2025).

Chain-of-Thought (CoT) is one of the most compelling phenomena discovered during the recent study of LLMs. In brief, it can be shown that eliciting a series of intermediate reasoning steps significantly improves the ability of LLMs to perform complex reasoning (Wei et al., 2022). The reasoning steps may be hierarchically structured themselves to break up involved problems into simpler subproblems (Yao et al., 2023; Bubeck et al., 2023).

**Conjecture 1** (Chain-of-Thought exploits Compositionally Sparse Functions). *Chain-of-Thought explicitly decomposes a compositionally sparse learning problem into sparse subproblems, each one of which can be learned. As such, it overcomes the complexity of one-shot learning.*

The following sketch shows how CoT fits into the compositional sparsity framework. Let $f_\theta : \mathbb{T}^d \to \mathbb{T}$ be a token-to-token predictor, $S[s : e]$ be the subsequence operator on a token sequence $S$ selecting all tokens from the $s$-th to the $e$-th token (inclusive), and $[\cdot, \dots]$ be the concatenation operator concatenating sequences of tokens. Then, using

CoT prediction can be understood as repeatedly applying the same predictor to a changing sequence of inputs.

$$f_\theta([\dots, f_\theta([X[2 : d], f_\theta([X[1 : d], f_\theta(X)])])]) \quad (5)$$

It is always possible to factor $f_\theta(\cdot)$ over a partition $\bigcup_{i \in I} P_i = \mathbb{T}^d$, s.t.,

$$f_\theta(x) = g_{i,\vartheta_i}(x) \quad \text{with } i \in I,\ x \in P_i \quad (6)$$

and thus understand $f_\theta(\cdot)$ to implicitly be a decision tree assigning each input to the leaf predictor $g_{i,\vartheta_i}(x)$ of the input's partition set (construction akin to Belcak & Wattenhofer (2023)). The plausibility of this idea is supported by, e.g., the work of Gan et al. (2024), who demonstrated that decision tree structures naturally emerge in auto-regressive language models. In the framework of compositional sparsity, each $g_{i,\vartheta_i}$ constitutes a simpler function of (bounded) density $c \ll d$. As such, the token-to-token predictor is faced with sparse, learnable functions $g_{i,\vartheta_i}$ of limited complexity and can solve the problem via repeated prediction. Note that decision trees themselves are sparse (Mansour, 1994), implying that the full function DAG is compositionally sparse. To directly predict the result in a single prediction step, i.e., evaluating $f_\theta$ on the input sequence $X$ once, on the other hand constitutes a dense problem and is therefore not (easily) learnable if at all.

However, there remain open questions, e.g., how exactly the subfunctions are represented, selected, and learned. Viewing this problem through the lens of compositional sparsity may be helpful in identifying promising research directions. The work of Cheung et al. (2019) shows how a single model can represent several functions using superposition and is thus possibly an answer to the question of representation.

### 4.4. Open Questions

Despite significant theoretical and empirical progress, several critical gaps remain in our understanding of how compositional sparsity interacts with DL in practice. We highlight below a set of open questions that, if addressed, could unify the theory of compositional sparsity with the practical realities of modern DL.

**Which Functions Are Learnable? And by which DNNs?**
A central challenge is to characterize the classes of compositional sparse functions – represented as DAGs – that can be efficiently inferred from data. While compositional sparsity provides a powerful representational framework, it remains unclear for which DAG topologies the underlying structure can be reliably discovered by learning algorithms. Notably, recent work suggests that certain function classes, such as staircase functions, may not require a strictly compositional structure, but rather an overlap in the subsets of variables they use. Careful analysis of these cases, as discussed in

the literature, may reveal subtler forms of compositionality relevant for learnability.

Optimization in the compositional sparsity framework suggests that the DAG of the target function must be reflected in the DNN's computational graph as a subgraph implying that only certain architectures can efficiently be optimized for certain target functions. This conjecture constitutes a direct pathway to empirical study of the relationship shared by the graph topologies of the target function and the network. Resulting insights could aid in delimiting the scope of our proposition and competing frameworks of understanding DNN optimization behavior.

**How Much Supervision Is Needed for Efficient Learning?** Another open question concerns the amount and type of supervision required to efficiently learn compositional functions. In practice, providing intermediate supervision – such as CoT steps or explicit labels for subproblems – can dramatically reduce the complexity of learning. However, it is not yet well understood how much and what kind of intermediate supervision is necessary to avoid exponential scaling in sample or computational complexity. Determining the minimal supervision needed for efficient learning remains an important direction, both for theory and practice.

**Why Are Multiple Layers Essential in Transformers?** The empirical success of multilayer transformers in pretraining tasks suggests that predicting the next word in natural language may, in general, require learning a compositionally sparse function that cannot be represented by a single threshold layer. This observation raises further questions about the depth and architectural requirements for capturing compositional structure in practice, and about the specific mechanisms by which transformers exploit or induce such sparsity.

**Can SGD Alone Discover Compositional Structure?** Recent work by Beneventano et al. (2024) shows that stochastic gradient descent (SGD) can naturally recover the support of the target function at the input layer. This raises a fundamental question: can deep networks, in practice, also discover the sparse constituent functions that would appear in a compositional decomposition of the target function? Is the implicit bias of SGD alone sufficient to induce this structure, or are additional biases—such as explicit sparsity constraints (e.g., $L_1$-regularization) or architectural priors—necessary to reliably recover compositional sparsity across layers? Addressing these questions is crucial for understanding the mechanisms by which deep networks learn and represent compositional structure.

**How Do Neural Networks Choose Among Multiple Decompositions?** Beyond these challenges, another subtle but important issue arises: a given function may admit many distinct compositional sparse decompositions, each corresponding to a different hierarchical arrangement of constituent functions. If a NN learns such a decomposition, which one does it select among the many possibilities? Are certain decompositions favored due to architectural inductive biases, optimization dynamics, or properties of the data? What determines this preference, and can it be controlled or predicted? Understanding the factors that bias the learning process toward particular decompositions is important for both interpretability and control.

Addressing these open questions is essential for bridging the gap between the theoretical foundations of compositional sparsity and the practical achievements of DL. Progress in these areas will not only deepen our understanding of why DNNs work so well, but may also guide the design of more efficient, interpretable, and robust learning systems.

## 5. Alternative Views

Compositional sparsity is not the only theory that serves as a candidate to explain approximation and optimization results for DNNs in high-dimensional learning problems. Manifold learning and hierarchical learning in the multi-index model have been studied as suitable alternatives.

**Manifold Learning** In the case of manifold learning, the data underlying a learning task is assumed to lie on or near a low-dimensional manifold embedded in the observable high-dimensional ambient space $\mathbb{R}^d$, the so-called *manifold assumption*. As such, the learning problem can be composed from an embedding function $g$ and a problem function $h$

$$f := h \circ g \qquad (7)$$
$$\text{where} \quad g : \mathbb{R}^d \to \mathcal{X}^k, \, h : \mathcal{X}^k \to \mathbb{R}$$

with $k \ll d$ and $\mathcal{X}$ not necessarily Euclidean. Representing and learning the problem function $h$ has now become trivially easy for sufficiently small values of the problem dimension $k$ and therefore avoids the curse of dimensionality. The case of the embedding function $g$, on the other hand, is more involved. A common assumption on $g$ posits that the mapping to the low-dimensional space is smooth and thus easy to learn. This condition is satisfied by synthetic data (e.g., S-curved manifold, Swiss-roll manifold, open box, torus, sphere, fishbowl, etc.). Real-world data manifolds often have varied properties that deviate from ideal assumptions (Kiani et al., 2024), so the effective dimensionality may be smaller than required to reasonably explain avoidance of the curse (Izenman, 2012). While the classical Nash embedding theorem proves that isometric embeddings into higher-dimensional spaces exist in theory, no known practical algorithm can reliably construct such an embedding. As pointed out by Meilă & Zhang (2023), the question

remains open whether we can efficiently learn mappings $g$ that preserve local geometric structure for complex, high-dimensional data.

In conclusion, manifold learning can serve as an explanation of why DNNs avoid the curse of dimensionality in approximation if the manifold assumption is true. If the mapping from the ambient space to the embedding space can be found efficiently, the explanation also extends to optimization. The recent work of Liang et al. (2024) hints at how the concepts of compositionality and manifold learning could also be considered as complementary concepts and may both be required to wholly explain the workings of DNNs.

**Multi-Index Model**   The multi-index model (Box & Cox, 1964; Bickel & Doksum, 1981) poses another avenue of studying high-dimensional learning problems (cf. Bruna & Hsu (2025) for a comprehensive survey). In this model, a regression function $f : \mathbb{R}^d \to \mathbb{R}$ can be expressed as the composition of a low-rank linear transformation $L$ and a link function $g$,

$$f(x) = g(Lx) \qquad (8)$$
$$\text{where} \quad L \in \mathbb{R}^{r \times d}, \ g : \mathbb{R}^r \to \mathbb{R}, \ r \ll d.$$

The row space $\mathrm{ran}(L^T)$ captures those (few) directions of the (high-dimensional) input space that are informative for predicting $y$. The optimization behavior of NNs can thus be interpreted as a two-phase process: In the first phase, the *search phase*, the model sifts through the noise and identifies the relevant input space components. Subsequently, during the *descent phase*, the model fits the target function on this set of relevant components. Notably, if the target function $f$ is a linear combination of several basis elements[1], the optimization follows a step-wise pattern in the online setting and the components are identified iteratively (Abbe et al., 2022; 2023). If these components are hierarchically structured, i.e., (statistically) hard-to-learn high-order components share input components with low-dimensional easy-to-learn components, learning the function is comparatively much faster than learning isolated high-order components. This property may be satisfied by real-world problems in which DNNs are successful.

However, existing works suffer from limitations: commonly, they consider shallow 2-layer, or heavily restricted 3-layer NNs, online, projected, and spherical Stochastic Gradient Descent (SGD), and layer-wise optimization to name a few practices seldom reflected in real-world training (Abbe et al., 2022; 2023; Arnaboldi et al., 2024a; Arous et al., 2021; Lee et al., 2024). As the work by Arnaboldi et al. (2024b) demon-

---

[1]A distribution over an input space induces a Fourier basis of orthogonal functions that are pair-wise uncorrelated under this distribution.

strates, seemingly inconsequential assumptions—such as not reusing data points—can entirely change the number of samples required and the speed at which entire function classes are learned. This showcases a need for further study to investigate which of the existing results extend to DNNs optimization with standard optimizers.

## 6. Discussion and Outlook

In this position paper we prove the conjecture that a large class of functions, namely all efficiently Turing-computable functions, share the property of being compositionally sparse. Theoretical findings on DL prove that it is this property of functions that allows DNNs to overcome the curse of dimensionality in relation to approximation. As such, they are universal approximators for all efficiently Turing-computable functions. On the end of optimization and generalization an abundance of empirical evidence underlines the capabilities of DL, but the theoretical results require strengthening. Notably, it is not entirely clear, when and how deep, potentially dense, learners manage to recover the sparse structure of the underlying task. One of the most prominent emergent phenomena of present day DL and LLM research, namely CoT reasoning, naturally fits into the proposed framework of compositional sparsity.

In conclusion, compositional sparsity represents a unifying principle:

- *Approximation:* It explains how DNNs can approximate complex tasks, namely, all polynomial-time computable functions without exponential blowup.

- *Optimization:* It suggests that discovering or revealing the "right" compositional DAG is the main difficulty, tackled in part by specialized architectures (CNNs) or new training paradigms (CoT, hierarchical prompting).

- *Generalization:* It enables smaller effective dimensionality thereby mitigating overfitting in practice.

Future work in DL theory will likely refine these ideas further, especially around the open-ended *optimization* challenge of how best to discover compositional structure from limited supervision. If we can more directly incorporate compositional assumptions (e.g., by guiding subfunction learning), we may see further breakthroughs in efficiency and interpretability.

### Acknowledgements

This work was partially supported by the Center for Brains, Minds, and Machines (CBMM) at MIT, NSF grant CCF-1231216, and other research sponsors. Davide D'Ascenzo was financially supported by the Italian National PhD Program in Artificial Intelligence (DM 351 intervento M4C1

- Inv. 4.1 - Ricerca PNRR), funded by NextGenerationEU (EU-NGEU). We thank many colleagues for discussions on these topics, especially members of the CBMM community.

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

## A. Proof of Theorem 3.2

**Theorem** (Efficient Computability implies Compositional Sparsity). *Any function $f \in$ **FP** is compositionally sparse.*

*Proof.* Assume a function $f \in$ **FP**, i.e., $f$ is efficiently Turing-computable. By definition, there exists a Deterministic Turing Machine (DTM) that computes $f$ in time $T(n) = \mathcal{O}(\text{poly}(n))$.

A DTM running in time $T(n)$ can be converted into a Boolean circuit with $\mathcal{O}(T(n) \log T(n))$ gates (Arora & Barak, 2009). For $T(n) = \mathcal{O}(\text{poly}\,n)$, this results in a circuit $C$ of size $\mathcal{O}(\text{poly}\,n)$.

The circuit $C$ may include gates with unbounded fan-in. To ensure compositional sparsity, we transform $C$ into an equivalent circuit with 2 inputs.

In particular:

- Any gate with $k > 2$ inputs can be replaced with a binary tree of $\lceil log_2(k) \rceil$-depth and $k - 1$ gates (see Figure 3 for an example visualization).

- Since $k$ in the original circuit is bounded by $\mathcal{O}(T(n)) = \mathcal{O}(\text{poly}(n))$, transforming $C$ into a circuit with fan-in 2 increases the circuit size by at most a polynomial factor. Thus, the final circuit has still polynomial size.

The final circuit is a Boolean circuit of $\mathcal{O}(\text{poly}(n))$ gates with fan-in 2. Since this circuit does not contain any cycles by construction, it can be translated into a DAG, where the Boolean inputs constitute the leaves, the gates are the internal nodes, and the output is the root. The circuit therefore computes a compositionally sparse function in the sense of Definition 3.1, which concludes the proof.

$\square$

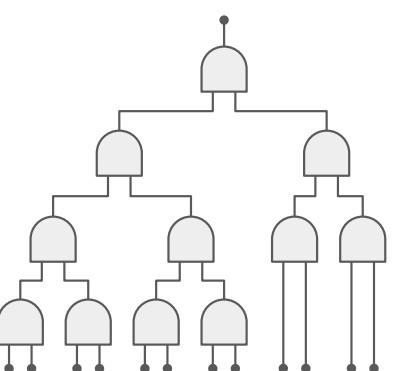

*Figure 3.* An AND gate with fan-in 12 decomposed in a tree of AND gates with fan-in 2.

