# OpenReview forum: "Position: A Theory of Deep Learning Must Include Compositional Sparsity"
_ICML.cc/2025/Position_Paper_Track — ICML 2025 Position Paper Track poster_

### Official Review · Reviewer_TLkw · 2025-02-28

**Significance:** 2
**Argument Clarity:** 1
**Rating:** 2
**Confidence:** 3

**Questions:**

- Concerning point (2): all the referenced papers seem to adopt very ad-hoc techniques to exploit the underlying sparsity (e.g., "layerwise stochastic coordinate descent"). What about pure SGD (or Adam), which is the default choice in practice?
- Also concerning point (2): for Transformers, even if a single attention layer is sparse (which is not true), the next one will process tokens that are weighted averages of all previous tokens. Hence, the CS assumption would be invalid? In addition, the theory would predict that fully sparse attention (e.g., entmax-based) would perform better, which contradicts empirical observations. Finally, what about recent recurrent models?
- Concerning CoT: can you clarify the points described above?
- "The work by Yau et al. (2024) bridges the gap from approximation to learning such with computations polynomially bounded histories." I do not understand this sentence.

**Discussion Potential:**

2

**Paper Summary:**

The paper considers the so-called "compositional sparsity" (CS) principle as a key component of a comprehensive deep learning theory. A function has CS if it can be decomposed in sub-functions, each of which acts on a subset of the original inputs. Thus, each CS function can be represented by a small DAG over the constituent subfunctions and the inputs. The paper considers three aspects:

1. On the learnability side, they prove a conjecture by Poggio & Fraser stating that CS functions fundamentally overlap with Turing-efficient functions and can be efficiently learned by a deep NN.

2. On the optimization side, they first review a serie of works on convergence rates for optimization with CS functions, and they then analyze existing NN models on the basis of this principle (in CNNs, CS is enforced explicitly by constraining the weight matrix to a block structure, while in Transformers a soft form of CS is enforced by the attention operator).

3. In the final part of the paper, they consider autoregression over CS functions, and they conjecture that the CS principle can help explaining the performance of chain-of-thought (CoT) models.

In the final part of the paper they overview two "opposing" views, namely the manifold hypothesis and the overparameterization regime.

## Update after rebuttal

Based on the answers, I am still under the impression that the paper is a strong contribution on the learnability side (due to the proof of the conjecture), but the broader position remains a bit vague and, when explicitly stated, does not seem unconventional. Also, the paper requires substantial rewriting. Thus, I retain my original vote.

**Position:**

No

**Position In Title:**

Yes

**Related Work:**

3

**Strengths And Weaknesses:**

I will overview in sequence the key sections of the paper to highlight the relative strengths and weaknesses.

1. The learnability section is the clearest section of the paper, but it is mostly a review of the material from Poggio & Fraser. I am not an expert on the topic, so I am unable to check the proof of the conjecture in detail. If true, it's a valid technical contribution but it does not add a lot to the "position" part of the paper.

2. By contrast, I have felt the optimization part weaker. If the CS principle helps explain the optimization and design of deep networks, I would have expected the paper to provide a clear set of testable hypotheses / ideas that could validate it, possibly open problems, etc. Instead, some of the material (esp. Section 4.2) appears circumstantial at best and lacking any falsifiable statement (see questions below).

3. To be honest I have found the third part unclear. Why I understand that CoT helps the model decomposes the output in multiple solvable subpieces, I do not understand how it connects to the previous discussion. In Eq. (5) it seems the model is applied autoregressively, each time removing one input token - however, this applies to any autoregression process, not just CoT (and it is also not valid in practice where the context is large enough to fit both the original input and the generated CoT).

Overall, while the topic is interesting, I do not feel the paper provides a cohesive view across the topics that can be framed as a single position. In addition, the position does not seem to provide any valid prediction or ideas for further designs (e.g., how can we learn or exploit the underlying compositional modularity?), limiting its potential scope and impact beyond the pure learnability (where the paper simply restates ideas from a previous paper).

**Support:**

2

---

> ### Author Rebuttal · Authors · 2025-03-31
>
> We thank the reviewer for their thoughtful feedback and constructive critiques.
>
> **Strengths And Weaknesses:**
> 1. *The learnability section is [...]*
>
>     We appreciate this observation. While the proof of Theorem 3.2 primarily establishes foundational credibility, its inclusion directly supports our position by bridging efficient computability to compositional sparsity (Corollary 3.4). The argument is that Poggio \& Fraser show how compositional sparsity of efficiently computable functions underlies approximation and generalization by deep (and sparse) neural networks. A natural question that follows is whether CS provides insights in optimization from data.
>
> 2. *By contrast, I have felt the optimization part weaker [...]*
>
>     This is a fair criticism. To avoid redundancy, we have addressed this in detail in our response to Reviewer a2RM (see Strengths and Weaknesses, point 2).
>
> 3. *To be honest I have found the third part unclear [...]*
>
>     We apologize for the lack of clarity that we will try to correct. The observation is that since efficiently Turing computable functions admit from coarse to very fine compositional representations (that is with high sparsity and high depth) CoT helps effectively providing training data under such decompositions. Results by Malach (2024) imply that  learnability is then much easier. In particular, LLMs are next-token predictors that exploit the sparse hierarchical structure of language (e.g., syntax trees) (see answer to Question (2)). They ignore however the latent "thinking steps" that take place between some of the words or some of the sentences.
>     Chain-of-Thought (CoT) addresses this by explicitly providing intermediate steps (e.g., "Socrates is human → humans are mortal"), revealing internal nodes of the reasoning DAG. We will clarify this and also include a figure.
>
> **Questions:**
> 1. *[...] What about pure SGD (or Adam), which is the default choice in practice?*
>
>     Thank you for raising this point. Recent work (Beneventano et al., 2024) shows that SGD can naturally recover the support of the target function. We hypothesize this occurs recursively across network layers, with each layer learning sparse, useful input combinations that collectively form an implicit DAG structure matching a compositional hierarchy.
>
> 2. *Also concerning point (2): for Transformers [...]*
>
>     While individual attention layers in Transformers do aggregate information across tokens, the compositional sparsity (CS) assumption applies to the hierarchical dependencies learned across layers. Each layer incrementally constructs higher-level abstractions (e.g., phrases, clauses) by combining sparse, local substructures from the previous layer. For instance, lower layers may capture syntactic dependencies (subject-verb agreement), while higher layers model semantic relationships (e.g., coreference). This aligns with findings that Transformers exhibit emergent compositional structure across layers (Murty et al., 2022; Vig \& Belinkov, 2019; Petty et al., 2024).
>     Moreover, a recent work (Deng et al., 2025) rigorously characterizes how sparse attention mechanisms in Transformers approximate exact attention, revealing that attention patterns naturally exhibit sparsity. Furthermore, in our own experiments we have found that the attentional maps in most layers of visual transformers are quite sparse. This is consistent with the observation that the receptive fields of units in trained transformers are similar to those of convolutional networks.
>
> 3. *In addition, the theory would predict that fully sparse attention (e.g., entmax-based) would perform better, which contradicts empirical observations.*
>
>     While we are not aware of any conclusive evidence that entmax-based attention performs worse than softmax, several studies have shown that sparse attention mechanisms can perform on par with or better than softmax in various tasks (Tezekbayev et al., 2021; Correia et al., 2019).
>
>     We would like to observe that enforcing sparsity in attention mechanisms should be done carefully as the enforced sparsity pattern should match the inherent structure of the task (i.e., as in CNNs for vision tasks). In language tasks, the optimal sparsity pattern may be less obvious and more dynamic.
>
> 4. *Finally, what about recent recurrent models?*
>
>     Assuming the question refers to Mamba-like architectures, we have not yet thoroughly investigated how these models might relate to compositional sparsity, we look forward to future work exploring these connections.
>
> 5. *The work by Yau et al. [...]*
>
>     We apologize for the oversight and thank you for the observation. The correct form of the sentence should have been "The work by Yau et al. (2024) bridges the gap between approximation and efficient learning of linear attention models, demonstrating that such learning can be achieved in polynomial time". We will make the correction.

---

> > ### Comment · Reviewer_TLkw · 2025-04-02
> >
> > **Strengths And Weaknesses:**
> >
> > 1. I agree, ok.
> > 2. Both research questions are, of course, valid, but they are also direct continuations of existing research. For a position paper, as I mentioned, I would have expected some unconventional research directions or hypotheses to test that would challenge the status quo. In addition, I believe adding these requires a substantial rewriting of the paper.
> > 3. While I thank the authors for the answer, I still find this point unclear. I will refrain from considering it further, but I acknowledge an additional rewriting is mentioned.
> >
> > **Questions:**
> >
> > 1. I believe this point should be addressed more in the paper, as the use of AdamW / SGD is fundamental in practice.
> > 2. Ok, I understand.
> > 3. Fair, I agree.
> > 4. Ok.
> > 5. Ok.
> >
> > Based on the answers, I am still under the impression that the paper is a strong contribution on the learnability side (due to the proof of the conjecture), but the broader position remains a bit vague and, when explicitly stated, does not seem unconventional. Also, the paper requires substantial rewriting. Thus, I retain my original vote for now.

---

### Official Review · Reviewer_a2RM · 2025-03-09

**Significance:** 4
**Argument Clarity:** 4
**Rating:** 4
**Confidence:** 4

**Questions:**

How do the works in the fields of compositional generalization, causal inference, and identifiability fit into your narrative?

**Discussion Potential:**

4

**Paper Summary:**

### Paper Summary

- Briefly summarize the paper, its contributions, and the position it advocates (if present). This summary should not be used to critique the paper. A well-written summary should not be disputed by the authors of the paper or other readers.

The authors argue that a key component for deep learning theory to explain the practical success of DNNs is *compositional sparsity*.
Compositional sparsity means that functions can be composed of "simple" composnents which rely on a small (thus, sparse) subset of the inputs.

The paper is structured as follows:
- Sec. 2: reviews classical learning theory and the curse of dimensionality
- Sec. 3.; it defines compositional sparisty and connects it to efficiecnt Turing-computability. Furthermore, the authors present related tehoretical results
- Sec. 4.; presents results of how compositional sparsity can enable learnability and sidestep worst case hardness issues. The authors also showcase how CNNs and Transformers display such sparse properties and provide universality results for autoregressive predictors and chain-of-thought. Finally, they pose concrete open questions.
- Sec. 5: discusses manifold learning as the alternative view, providing evidence from the literature that it might actually be a complementary point to compositional sparsity. Finally, the authors highlight the limitations of classical learning theory.
- Sec 6: concludes the paper, summarizing the three aspects (approximation, learnability, and generalization) that are unified by compositional sparsity

**Position:**

Yes

**Position In Title:**

Yes

**Related Work:**

2

**Strengths And Weaknesses:**

## Strengths
The paper is clearly written, the position and argumentation clear, the topic timely and relevant, the alternative views are represented. The position also relates to most recent advances (autoregressive models, chain-of-thought). It is sufficiently high-level for a position paper, though it does not lack detail.

## Weaknesses
- I could not find the definition of "efficient Turing-computability"
- The paper does provide two open questions, though it does not discuss concrete research questions worth exploring. Especially given the available place, I encourage the authors to use that space to provide clear guidance to the community about the most important/interesting research questions.
- **The related works miss a large body of work in compositional generalization, causal inference, and identifiability**, which often also rely on sparsity; see the references in these works:
	- compositional generalization: https://arxiv.org/abs/2411.07784
	- causal inference/Independent Causal Mechanisms/Sparse Mechanism Shift hypothesis: http://arxiv.org/abs/2206.02013, https://www.tandfonline.com/doi/full/10.1080/00949655.2018.1505197
	- identifiability: http://arxiv.org/abs/2107.10098

If these points are addressed, I am reconsidering my score.

**Support:**

3

---

> ### Author Rebuttal · Authors · 2025-03-31
>
> We thank the reviewer for their thoughtful feedback and constructive critiques.
>
> **Strengths And Weaknesses:**
> 1. *I could not find the definition of "efficient Turing-computability"*
>
>     We appreciate the opportunity to clarify this terminology. The term "efficient Turing-computability" was intentionally left loosely defined in the original text to avoid overcomplicating the exposition, but we will revise the main text to explicitly ground this concept in the complexity class FP. We acknowledge, however, that computability for real-valued functions can be formalized in multiple ways (for a detailed discussion, see Poggio and Fraser, 2024, *Compositional sparsity of learnable functions*). In the context of Theorem 3.2, the relevant framework is the complexity class FP, which consists of function problems solvable by a deterministic Turing machine in polynomial time. Whereas the decision problem class P involves solutions that are single-bit yes/no answers, FP represents functions with outputs computable in polynomial time. Although the nuances of real-number computability are beyond the scope of this work, we equate "efficient Turing-computable functions" with FP for simplicity.
>
> 2. *The paper does provide two open questions, though it does not discuss concrete research questions worth exploring. Especially given the available place, I encourage the authors to use that space to provide clear guidance to the community about the most important/interesting research questions.*
>
>     We appreciate the suggestion. As we mentioned in our response to Reviewer TLkw (Strengths And Weaknesses, point 1), a key open question is indeed the role of CS in optimization. Our main goal was to formulate this question for the community. However, following your suggestion, we will be more specific on how this open problem may be addressed. In particular, we will state the following more specific question: *Under what structural assumptions on the DAG can compositional sparsity enable polynomial-time learnability?* We will suggest that this research challenge aligns with recent efforts on optimization of staircase functions (Abbe et al., 2021, *The Staircase Property*) and of sparse Boolean polynomials (Negahban \& Shah, 2012, *Learning Sparse Boolean Polynomials*). We will also reference the role of gradient-based methods in implicitly recovering the support of target functions (Beneventano et al., 2024, *How Neural Networks Learn the Support is an Implicit Regularization Effect of SGD*), connecting to SGD's empirical success (see also answer to Reviewer TLkw, Question 1). In both cases a specific form of compositional sparsity is needed to guarantee optimization.
>
> 3. *The related works miss a large body of work in compositional generalization, causal inference, and identifiability, which often also rely on sparsity*
>
>     We appreciate your reference to compositional generalization, causal inference, and identifiability literature. These fields emphasize structural sparsity, which aligns with our work. We will update our related work section to draw these connections and cite relevant works.

---

> > ### Comment · Reviewer_a2RM · 2025-04-02
> >
> > Thank you for your detailed response. I updated my score to 4.

---

### Official Review · Reviewer_as1F · 2025-03-12

**Significance:** 4
**Argument Clarity:** 4
**Rating:** 4
**Confidence:** 3

**Questions:**

Given that the authors are discussing how to partition the input space and how to represent general functions/algorithms as DAGs, there should be citation and discussion of

Smale's original work "On the Topology of Algorithms"

as well as Michael Shub's review paper "On the Work of Steve Smale on the Theory of Computation"

These works are essentially the foundation of modern Topological Complexity Theory, which, with the recent introduction of Topological Deep Learning, could be a fruitful interaction area for your suggested area of research.

**Discussion Potential:**

2

**Paper Summary:**

The paper begins with a review of the major questions of deep learning: approximation, optimization, and generalization. It then proceeds to review the famed "curse of dimensionality" which comes out of statistical learning theory. Essentially learning a (Lipschitz) function $f:\mathbb{R}^d \to \mathbb{R}$ up to $\epsilon$ accuracy requires $\mathcal{O}(\epsilon^{-d})$ samples, which is exponential in $d$.

Research by Beneventano, et al (2021) argues that Deep Neural Networks (DNNs) effectively learn such functions with far fewer samples, which suggests that DNNs don't suffer from the curse of dimensionality.

The authors argue that this is because most functions of interest are actually compositionally sparse, this means that it can be rewritten as a composition of polynomially-many (poly in $d$) functions that each depend on only a fixed number $c<< d$ number of coordinates. The authors support their claim by proving a conjecture from Poggio & Fraser (2024) stating that all efficiently Turing computable functions are compositionally sparse. The main argument for this theorem is given in the appendix, which converts a Deterministic Turning Machine into a Boolean circuit, which has the structure of a directed acyclic graph (DAG). This DAG is essentially the composition tree for the DNN representation of the function.

The paper then proceeds to outline some theoretical challenges, such as the fact that learning this composition tree, without partial information is NP-complete, so computationally infeasible in practice. This is, again, at tension with the success of DNNs in practice. This motivates two open questions:

1. Which DAGs (composition trees) are learnable? and

2. What is the minimal amount of supervision (or partial information) needed for a DNN to infer the composition tree?

Speculation on why certain NN architectures seem to succeed, e.g. CNNs exploiting translation invariance and transformers tuning attention in a way that partitions the space of inputs via autoregression. is given, as well as some alternative viewpoints from manifold learning.

## Update after Rebuttal

I think it's a very strong paper and deserves to be accepted. It's not the best position paper I've evaluated, but it's close! I think the proof of the conjecture is very significant and deserves to be published.

**Position:**

Yes

**Position In Title:**

Yes

**Related Work:**

3

**Strengths And Weaknesses:**

The main strength of the paper is it's mathematical clarity and succinctness. The paper is only 6 pages, which is a pleasure. The result, if true, seems to be a major step forward in the understanding of why DNNs are not affected by the curse of dimensionality.

The main weakness is that the paper reads more like a research paper, which is at odds with the position track. This might also mean there is less spirited discussion.

There is also a very minor weakness in how the abstract is written: I would break up the first sentence of the abstract into two clearer, shorter sentences. The comma before "that drives their success" is unnecessary, creating an awkward pause. Delete the comma and "that" so it has more of an active voice.

**Support:**

4

---

> ### Author Rebuttal · Authors · 2025-03-31
>
> We appreciate your positive feedback regarding the mathematical clarity and succinctness of our paper.
>
> **Strengths And Weaknesses:**
>
> 1. *The main weakness is that the paper reads more like a research paper, which is at odds with the position track. This might also mean there is less spirited discussion.*
>
>     We appreciate the suggestion. We will add more explicit open questions and conjectures, as highlighted in our response to  Reviewer a2RM (see Strengths and Weaknesses, point 2).
>
> 2. *There is also a very minor weakness in how the abstract is written: I would break up the first sentence of the abstract into two clearer, shorter sentences. The comma before "that drives their success" is unnecessary, creating an awkward pause. Delete the comma and "that" so it has more of an active voice.*
>
>      We appreciate the suggestion. We will revise our abstract in light of it.
>
>
> **Questions:**
> 1. *Given that the authors are discussing how to partition the input space and how to represent general functions/algorithms as DAGs, there should be citation and discussion of Smale's original work "On the Topology of Algorithms" as well as Michael Shub's review paper "On the Work of Steve Smale on the Theory of Computation". These works are essentially the foundation of modern Topological Complexity Theory, which, with the recent introduction of Topological Deep Learning, could be a fruitful interaction area for your suggested area of research.*
>
>     We appreciate your reference to Smale's "On the Topology of Algorithms" and Shub's review of Smale's work on computation theory. Both provide topological perspectives on algorithmic complexity that align with our discussed concept of compositional sparsity.
>
>     Smale and Shub demonstrate how topological constraints on the algorithm structure can shape computational complexity. Similarly, our work shows how hierarchical / DAG-based decomposition can factor high-dimensional complexity into simpler, low-dimensional components.
>
>     We will try to connect topological complexity theory to our framework, so we can highlight how both topological methods and sparse compositional architectures explain why deep networks can effectively avoid the curse of dimensionality.

---

> > ### Comment · Reviewer_as1F · 2025-04-05
> >
> > thank you for the comment, i look forward to reading the revision.

---

### Official Review · Reviewer_4ssu · 2025-03-14

**Significance:** 3
**Argument Clarity:** 3
**Rating:** 4
**Confidence:** 4

**Questions:**

Please see the above weakness.

**Discussion Potential:**

4

**Paper Summary:**

This paper argues that the effectiveness of deep learning may be coming from compositional sparsity in the structure, and this must be an important new direction in understanding it. Based on the recent findings regarding the approximation complexity of deep learning and the compositional sparsity of Turing computable functions, this paper combines all these findings and points out that the recent deep learning methodologies (especially CNN, auto-regressive predictors, Chain-of-Thought, and partially Transformer) may be in the intersection of these concepts. The paper also outlines unresolved questions in this direction, and suggests some research directions.

## update after rebuttal

All reviewers have agreed on the value of the paper. I maintain my original score.

**Position:**

Yes

**Position In Title:**

Yes

**Related Work:**

4

**Strengths And Weaknesses:**

+ This paper clearly shows the evidence that the effectiveness of recent deep learning might be coming from compositional sparsity, based on recent theoretical findings. I have checked the proof of Theorem 3.2, which is the key to connect the compositional sparsity assumption to the domain of practical deep learning, and I believe it is correct.

+ This argument is beneficial for the research community in that it pinpoints important directions for neural architecture design. Of course, it has already been discussed for a long time that inductive biases are essential for the success of deep learning, but this paper articulates the concept into compositional sparsity. This can give more clarity on where we should look.

- One downside of the paper I think is that the paper is largely based on existing works. The paper is about connecting these dots. I still find this paper to be a good position paper, but I also think that some people might say that this paper is merely repeating the conclusions other papers have already found.

- I do not see any other particular weaknesses.

**Support:**

3

---

> ### Author Rebuttal · Authors · 2025-03-31
>
> Thank you for your positive review.

---

### Decision · Program_Chairs · 2025-04-30

**Decision:**

Accept (poster)

**Comment:**

This work puts forth a clear hypothesis underlying the efficiency in training and learning of deep neural networks. This hypothesis is backed up with some theoretical foundations, and some future directions are proposed.

Reviewers agree that more concrete conjectures or consequences of making progress towards this hypothesis may help into making this paper more useful, as well as there is need to improve the presentation quality. Upon these conditions, I am in favor of acceptance.